# Ultra-rapid and highly efficient enrichment of organic pollutants via magnetic mesoporous nanosponge for ultrasensitive nanosensors

Lingling Zhang [1,5], Yu Guo[1,5], Rui Hao[1], Yafei Shi[1], Hongjun You [2], Hu Nan[3], Yanzhu Dai[3], Danjun Liu[4], Dangyuan Lei [4] & Jixiang Fang [1,2✉]

Currently, owing to the single-molecule-level sensitivity and highly informative spectroscopic characteristics, surface-enhanced Raman scattering (SERS) is regarded as the most direct and effective detection technique. However, SERS still faces several challenges in its practical applications, such as the complex matrix interferences, and low sensitivity to the molecules of intrinsic small cross-sections or weak affinity to the surface of metals. Here, we show an enrichment-typed sensing strategy with both excellent selectivity and ultrahigh detection sensitivity based on a powerful porous composite material, called mesoporous nanosponge. The nanosponge consists of porous β-cyclodextrin polymers immobilized with magnetic NPs, demonstrating remarkable capability of effective and fast removal of organic micropollutants, e.g., ~90% removal efficiency within ~1 min, and an enrichment factor up to ~$10^3$. By means of this current enrichment strategy, the limit of detection for typical organic pollutants can be significantly improved by 2~3 orders of magnitude. Consequently, the current enrichment strategy is proved to be applicable in a variety of fields for portable and fast detection, such as Raman and fluorescent sensing.

[1] Key Laboratory of Physical Electronics and Devices of Ministry of Education, School of Electronic Science and Engineering, Faculty of Electronic and Information Engineering, Xi'an Jiaotong University, Xi'an, Shaanxi 710049, China. [2] Key Laboratory of Biomedical Information Engineering of Ministry of Education, School of Life Science and Technology, Xi'an Jiaotong University, Xi'an, Shaanxi 710049, China. [3] School of Microelectronics, Faculty of Electronic and Information Engineering, Xi'an Jiaotong University, Xi'an Jiaotong University, Xi'an, Shaanxi 710049, China. [4] Department of Materials Science and Engineering, City University of Hong Kong, 83 Tat Chee Avenue, Kowloon 999077 Hong Kong, China. [5]These authors contributed equally: Lingling Zhang, Yu Guo. ✉email: jxfang@mail.xjtu.edu.cn

The Stockholm Convention on Persistent Organic Pollutants (POPs) was endorsed by 131 nations in 2004 to eliminate the most persistent bioaccumulative and toxic substances in the world[1]. Organic micropollutants of ground and surface water resources, such as pesticides and plastic components, have aroused great concerns about potential negative effects on aquatic ecosystems and human health[2,3]. Therefore, parallel to the researches of adsorbent materials to remove organic pollutants from water, the ultrasensitive detection of organic pollutants is another crucial field, since the solubility of organic micropollutants in water is always at the trace level[4]. Among diverse detection approaches, surface-enhanced Raman spectroscopy (SERS), achieved breakthroughs in 1997 and became the first vibrational spectroscopy technique that could provide delicate information on molecular fingerprints with a potential of single-molecule sensitivity[5–9], thus regarded as the most simple, fast, flexible, and portable detection technique.

However, it seems surprising that after fifty decades, SERS has not yet been widely used in practical applications[10]. This is owing to the fact that, besides the stability of SERS substrates and reproducibility of spot-to-spot, SERS still faces two major bottlenecks in the commercial market. The first is the low detection sensitivity to the molecules of intrinsic small cross-sections or weak affinity to metal surface. As we know, SERS is an optical near-field effect[11–13]. A high activity can be obtained only when the target molecule is very close to the plasmonic surface. Therefore, researchers exploited diverse approaches to capture the target molecules onto the metal surface by means of antibodies, aptamers, ion liangs, et al.[14,15]. However, the ultra-rapid capture to meet on-site and portable detection remains a challenge. The second is the interference from the complex matrices[16]. In real-sample detection, most organic pollutants in water cannot be effectively adsorbed onto the metallic surface because of their low affinity toward the metal, hence the metal surface is usually inactivated due to unspecific adsorption by the interference from matrix molecules in the surrounding environment. Thus, for commercial applications, an ultra-fast and effective pretreatment is of importance to eliminate the most matrix interference. Therefore, recently, some strategies, e.g., selective separation, concentration, enrichment from complex matrix, and spatial localization of target molecules[17–19], are suggested to solve this long-standing challenge.

In this work, we propose a new sensing strategy in rapid separation and highly efficient enrichment of POPs from complicated real-sample matrices by means of the magnetic NPs immobilized porous β-CD polymer (MN-PCDP), called mesoporous nanosponge. The current strategy (the schematic description of the protocol is shown in Fig. 1) demonstrates several remarkable advantages. Firstly, specific and selective absorption and separation of target molecules eliminate the matrix interference. When the MN-PCDP adsorbent is dispersed into the water in the beaker containing organic pollutants and impurities (shown in Fig. 1a, b), specific and selective adsorption of target molecules can be achieved. In fact, microporous β-CD material has been widely studied because of its outstanding adsorption efficiency through forming host-guest inclusion with many hydrophobic organic pollutants[20,21]. The magnetic NPs are introduced into the MN-PCDP compounds to rapidly separate the adsorbent from water, thus eliminates the interference of unspecific adsorption from the complex matrix in the real-samples. Secondly, highly efficient concentration and enrichment capability thus ultra-high detection sensitivity can be obtained. The MN-PCDP adsorbent demonstrates a remarkable removal efficiency on organic micropollutants, e.g., ~90% (relative standard deviation, RSD < 1%). Meanwhile, the adsorbed pollutants from initial water of ~1000 ml can be desorbed in ethanol with a volume of ~1 ml (Fig. 1c), for further analysis such as UV-vis, Raman, and fluorescent spectroscopy. Thus, an ultra-high enrichment efficiency with an enrichment factor up to ~$10^3$ times can be obtained (RSD < 5%), and the limit of detection (LOD) in a variety of sensing applications can be lowered by 2−3 orders of magnitude. Thirdly, ultra-quick enrichment processes thus on-site portable detection can be realized. In the current strategy, ultra-fast adsorption, magnetic separation, and desorption can be accomplished, i.e., totally within 2−3 min. Thus, the current sensing strategy can be believed to be applicable to a wider range of sensing areas for an economical, simple, fast, flexible, and portable detection.

## Results

**Synthesis and characterization of MN-PCDP nanosponges.** The microporous MN-PCDP material, an inexpensive and renewable carbohydrate, which is featured by small pores and high surface areas, is used in this work as an excellent adsorbent. The MN-PCDP is prepared by cross-linking polymerization of β-CD and cross-linking agent (tetrafluoroterephthalonitrile (TFT)), with magnetic NPs (Fe$_3$O$_4$) in one-step solvothermal reaction. Supplementary Fig. 1a−c shows the transmission electron microscope (TEM) images of magnetic NPs (MN, Fe$_3$O$_4$), porous β-CD polymer (PCDP), and MN-PCDP, respectively. As shown in Supplementary Fig. 1a, the synthesized MN exhibits regular spheres with good dispersibility and uniform size (average size ~200 nm). Supplementary Fig. 1b exhibits that the PCDP is a porous network structure. After the immobilization of MN, as shown in Supplementary Fig. 1c, the porous network structure of MN-PCDP is not disrupted. The Fourier transform-infrared

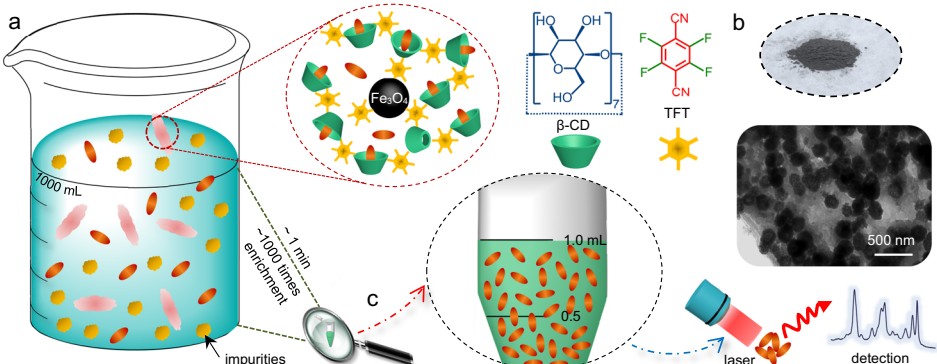

**Fig. 1 Schematic of the current enrichment and detection based on the porous β-CD polymer. a** Adsorption and **c** desorption processes using magnetic NPs immobilized porous β-CD polymer (MN-PCDP) with ~1000 times enrichment. **b** Optical photograph and TEM imagine of MN-PCDP.

spectroscopy (FT-IR) spectra of MN, β-CD, TFT, PCDP, MN-PCDP are displayed in Supplementary Fig. 1d. The absorption bands at 1652 and 1396 cm$^{-1}$ of the MN can be associated with carboxylate group[22]. The FT-IR spectrum of the MN-PCDP obviously combines the characteristic peaks of the TFT and the β-CD. The signal intensity of absorption peak at 1265 cm$^{-1}$ in relation to C-F stretching vibration is weaker than that in TFT owing to the partial replacement of F[23,24], implying that the β-CD has been crosslinked with TFT (Supplementary Fig. 2). Supplementary Figure 1e indicates that the Brunauer−Emmett−Teller surface areas ($S_{BET}$) of MN-PCDP is about 66 m$^2$ g$^{-1}$. The pores with diameter of 1.7−3.0 nm comprise the majority of the free volume of MN-PCDP and its average pore diameter is ~2.12 nm.

**Adsorption behavior of MN-PCDP nanosponges**. The high surface area and permanent porosity of MN-PCDP mesoporous nanosponge enable the rapid removal of organic micropollutants from water[25]. As shown in Supplementary Fig. 3, the PCDP and MN-PCDP display almost the same properties in time-dependent adsorptions of bisphenol A (BPA), revealing the immobilization of magnetic NPs has no remarkable influence on the adsorption performance of PCDP. The time-dependent adsorptions of various organic micropollutants adsorbed by MN-PCDP, including plastic components, pesticide, and aromatic model compounds (Fig. 2a), are shown in Fig. 2b, Supplementary Fig. 4, and Supplementary Table 1. Each organic micropollutant is rapidly removed, reaching ~95% of its equilibrium uptake in 10 s[20]. The removal efficiencies of BPA, parathion, carbendazim, and 2-naphthol (2-NO) by MN-PCDP are more than 80% in 30 s, which is much higher than the Norit ROW 0.8 supra extruded activated carbon (NAC) as presented in Fig. 2c, Supplementary Fig. 5, 6 and Supplementary Table 2. In this work, different cross-linking agent, e.g., epichlorohydrin (EPI) is compared

(Supplementary Fig. 7). As shown in Supplementary Fig. 8, the removal efficiency of BPA by MNEPI-CDP in 1 min is 19.5%, which is much lower than MN-PCDP. We further probe the readily accessible binding sites of MN-PCDP by determining the flow-through uptake of different organic micropollutants. In these experiments, the adsorbent (~5 mg) is trapped as a thin layer on a 0.22 μm syringe filter, and aqueous organic pollutants (5 mL, 0.1 mM) passed rapidly through the filter at a flow rate of 10 ml min$^{-1}$ (Supplementary Fig. 9). Under these conditions, for example, 76% of the BPA is removed from the solution, corresponding to more than 84% of its equilibrium adsorption, confirming that the host−guest interaction plays a major role in the filtration process by syringe[26].

As is known, the hydroxyl groups of β-CD are located at the outer surface of the molecule, that is, primary hydroxyls at the narrow side and secondary hydroxyls at the wider side, which makes β-CD water−soluble but simultaneously generates an inner cavity that is relatively hydrophobic[27]. Because of their hydrophobic interior cavity, β-CD can either partially or entirely accommodate suitably sized lipophilic low molecular weight molecules or even polymers[28]. The superior performance of MN-PCDP can be helpful to that its β-CD moieties are easily accessed by most of the organic micropollutants, and these molecules can be rapidly trapped in the cavity of β-CD. For example, MN-PCDP exhibits a remarkable adsorption capability and selectivity for most aromatics and some chain compounds, as shown in Fig. 2b and Supplementary Figs. 10, 11. Furthermore, by means of particular treatments such as changing pH value of solution[29], the adsorption feature of molecules can be tuned. Thus, the MN-PCDP mesoporous nanosponge will display a wide applicability and selectivity in a variety of molecules.

Furthermore, the influence of the concentrations of adsorbent on the adsorption efficiency of BPA is studied and shown in

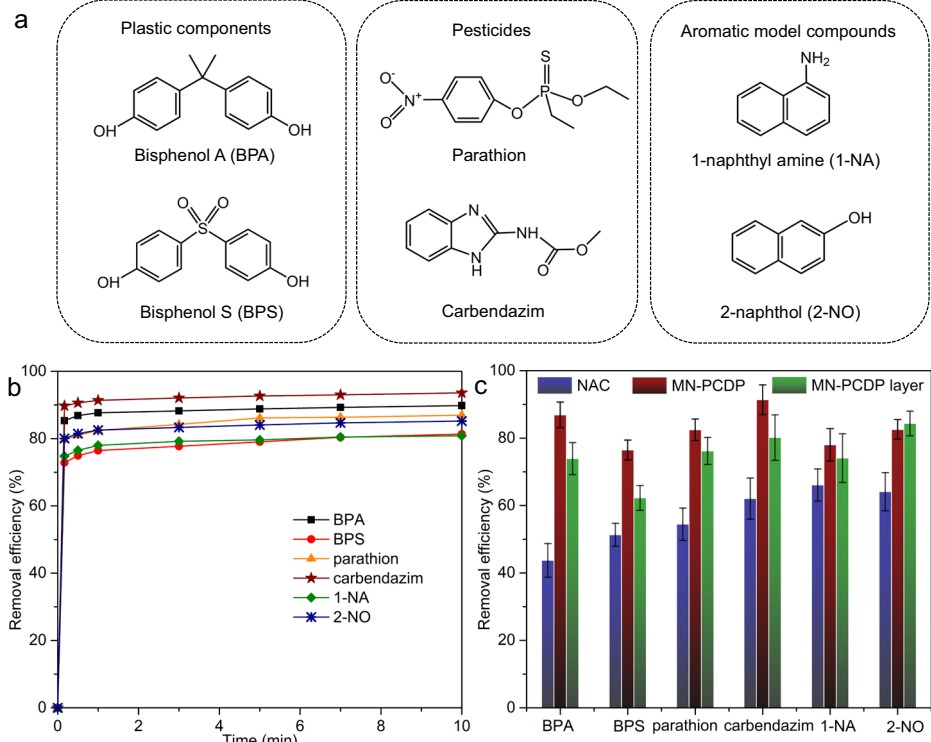

**Fig. 2 MN-PCDP rapidly adsorbs a broad range of organic pollutants. a** Structures of each tested organic pollutant. **b** Time-dependent adsorption of each pollutant (0.1 mM) by MN-PCDP (1 mg mL$^{-1}$). **c** Percentage removal efficiency of each pollutant obtained by stirring NAC (blue), stirring MN-PCDP (red), and rapidly flowing through a thin MN-PCDP layer (green). The data are reported as the average uptake of triplicate experiments. Error bars mean standard deviations.

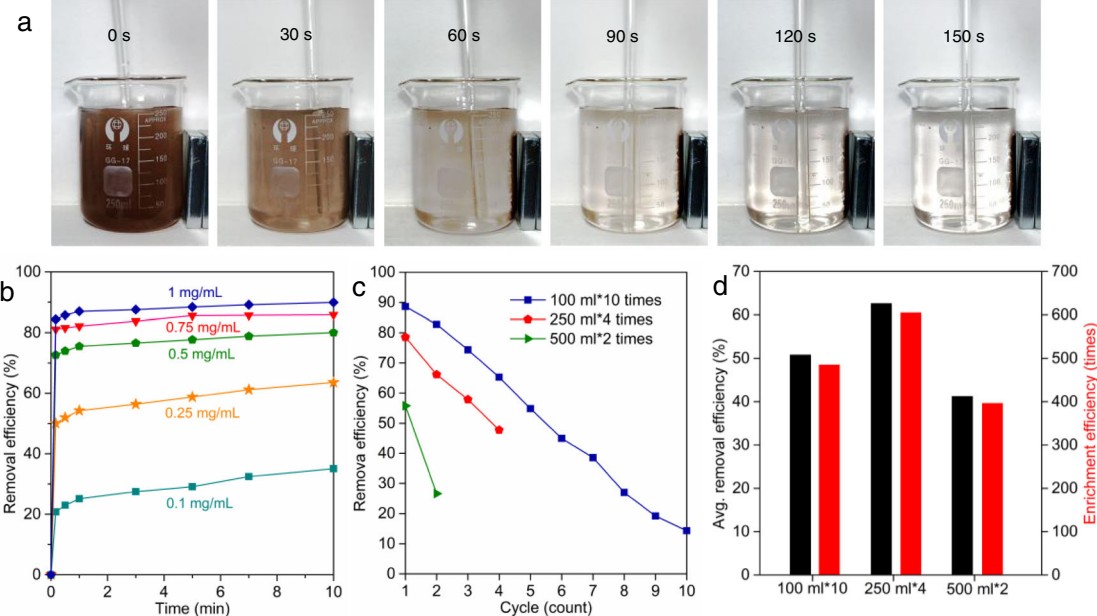

**Fig. 3 Rapid enrichment performance of MN-PCDP. a** Optical photographs of MN-PCDP separation process by a magnet in continuous time. **b** Time-dependent adsorption of BPA (0.1 mM) using MN-PCDP with different dosage (0.1, 0.25, 0.5, 0.75, and 1 mg L$^{-1}$). **c** Removal efficiency of BPA (0.01 mM) using MN-PCDP (100 mg) in three methods (100 mL for 10 times, 250 mL for 4 times and 500 mL for 2 times). **d** Average removal (black) and enrichment (red) efficiency of the three methods in (**c**).

Fig. 3b, Supplementary Fig. 12, and Supplementary Table 3. When the concentration increases from 0.1 to 1.0 mg L$^{-1}$, the adsorption efficiency of BPA is enhanced from 25.12 to 87.09% within 1 min and from 35.07 to 89.82% within 10 min.

**Desorption, concentration, and enrichment features of MN-PCDP nanosponges**. As we all know, organic micropollutants exhibit good solubility in organic solution, such as ethanol and methanol[20]. Hence, after adsorption process, we may separate the MN-PCDP mesoporous nanosponges from the solution quickly by the magnet, and utilize ethanol to desorb the organic micropollutants from the adsorbent, thus obtain the enriched pollutant solution. In order to obtain higher concentration of desorbed micropollutant solution, in this work, we chose 1 mL ethanol to desorb organic micropollutants adsorbed in MN-PCDP adsorbent. As shown in Supplementary Fig. 13, the concentration of BPA can be increased to 88.5 times of its initial concentration with a recipe of 100 mL organic pollutant (BPA) solution and 100 mg MN-PCDP adsorbent. This result reveals that more than 98% of the adsorbed organic micropollutants are desorbed into the ethanol solution. As discussed in Fig. 3b, with the concentration of adsorbent increases, the adsorption efficiency tends to reach equilibrium. Considering the cost increase of sample preparation and dosage of adsorbent in the desorption process (with 1 mL ethanol), 20−100 mg/100 mL of adsorbent is selected as the adsorbent concentration in subsequent experiments.

In order to further improve the enrichment efficiency of 100 mg adsorbent in total 1000 mL organic micropollutants, herein, we attempted three methods during the adsorption and desorption processes, including 100 mL × 10 times, 250 mL × 4 times, and 500 mL × 2 times. Importantly, the adsorbent can be simply separated by a magnet in every adsorption cycle, then desorbed in ethanol. As shown in Fig. 3c, Supplementary Fig. 14, and Supplementary Table 4, as the recycling adsorption times increase, the removal efficiencies of these three methods gradually decrease. The average removal efficiencies in methods of 100 mL × 10 times, 250 mL × 4 times, and 500 mL × 2 times are 50.98,

62.58, and 41.22%, respectively. These results represent an enrichment capability of 485, 605, and 396 times of the initial concentration (Fig. 3d), respectively. Thus, we achieve a notable enrichment factor of above 600 times of initial organic pollutants via 1000 mL initial solution by means of the optimization of adsorption and desorption processes. Here, the optimized parameters, i.e., 100 mg adsorbent in 250 mL × 4 cycle times, are used for the succedent experiments. Meanwhile, it is also worth pointing out that the separation process by a magnet is very fast and facile, such as 100 mL with 60 s (Supplementary Fig. 15a), 250 mL with 90 s (Fig. 3a and Supplementary Fig. 15b), and 500 mL with 150 s (Supplementary Fig. 15c). Therefore, the current ultra-fast enrichment protocol may fully meet the requirement of on-site and portable detection applications.

**SERS and fluorescence measurements of pollutants via current enrichment protocol**. In order to evaluate the advantage of the current enrichment protocol on the detection sensitivity, fluorescence, and SERS spectra of five POPs molecules, e.g., BPA, carbendazim, tetramethyl thiuram disulfide (TMTD, thiram), diquat, and anthracene, are measured using MN-PCDP nanosponges as adsorbent. The hydrophobic slippery SERS platform[30] with ~55 nm Au NPs (Supplementary Fig. 16) is adopted for the measurement of SERS spectra. As shown in Fig. 4a, b, without the enrichment process, the LOD of SERS for TMTD is around 1 pM. However, after the enrichment using MN-PCDP adsorbent, this value reaches to ~5 fM, showing an increase of $10^2−10^3$. Meanwhile, based on this enrichment protocol, the LOD of BPA, carbendazim, diquat and anthracene are up to 0.1 nM, 5 pM, 1 pM, and 1 nM (Supplementary Figs. 17−20), which are much lower than most of the magnetic SERS-based sensors (Table 1). Furthermore, multiple adsorption and desorption experiments by MN-PCDP for the above five organic molecules are implemented to illustrate the reproducibility of this adsorbent. In Supplementary Figs. 21−23 and Supplementary Tables 5, 6, the removal efficiencies and enrichment efficiencies of MN-PCDP adsorbent are excellent for target molecules with RSD less than 1 and 5%,

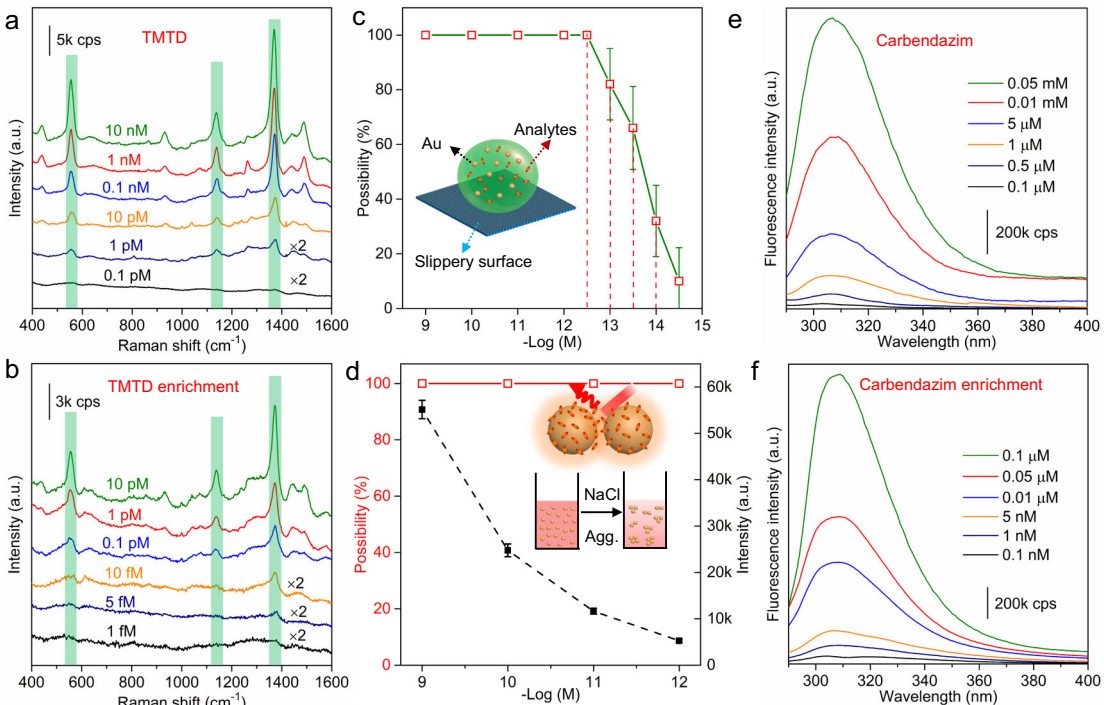

**Fig. 4 Application in Raman and fluorescence detection by this enrichment strategy.** Enhanced Raman spectra of TMTD **a** before and **b** after enrichment process of MN-PCDP. **c** Probability of SERS signals with different concentrations of TMTD after enrichment process of MN-PCDP by hydrophobic slippery SERS platform. The inserted schematic diagram shows hydrophobic slippery SERS platform. The data are reported as the average enrichment of five experiments. Error bars mean standard deviations. Shaded areas mean the characteristic peaks (555, 1138, and 1370 $cm^{-1}$) of TMTD molecule. **d** Detection probability (red) and intensity of SERS signals (black) with different concentrations of TMTD after enrichment process of MN-PCDP by aggregating approach with Au colloid. The inserted schematic diagram shows aggregating approach with Au colloid. The data are reported as the average enrichment of five experiments. Error bars mean standard deviations. Fluorescence spectra of carbendazim **e** before and **f** after the enrichment of MN-PCDP.

**Table 1 Detectability of the MNPs for organic pollutants reported in the literature.**

| Types of MNPs[a] | Analyte | LOD | Ref. |
|---|---|---|---|
| MN-PCDP[b] | TMTD | $5 \times 10^{-15}$ M | This work |
| $Fe_3O_4@SiO_2@Ag$ nanospindles | TMTD | $10^{-7}$ M | 43 |
| $Fe_3O_4@Au$ NRs array | TMTD | $10^{-9}$ M | 44 |
| $Fe_3O_4@SiO_2@Ag$ core-shell MNPs | TMTD | $10^{-9}$ M | 45 |
| Cube-like $Fe_3O_4@SiO_2@Au@Ag$ | TMTD | $5 \times 10^{-11}$ M | 46 |
| Flower-like $Fe_3O_4@SiO_2@Ag$ | TMTD | $10^{-11}$ M | 47 |
| $Fe_3O_4@Ag$-PEI-Au@Ag | TMTD | $5 \times 10^{-12}$ M | 48 |
| $Au@Fe_3O_4$ network | TMTD | $5 \times 10^{-14}$ M | 49 |
| MN-PCDP | Diquat | $10^{-12}$ M | This work |
| $Fe_3O_4@AuNRs$ assemblies | Diquat | $10^{-9}$ M | 44 |
| $Fe_3O_4@Ag$-PEI-Au@Ag | Paraquat | $10^{-10}$ M | 48 |
| MN-PCDP | Carbendazim | $5 \times 10^{-12}$ M | This work |
| $Au/Fe_3O_4$ nanocomposite | Carbendazim | $2.3 \times 10^{-9}$ M | 50 |
| MN-PCDP | BPA | $10^{-10}$ M | This work |
| Magnetic gold nanoclusters | BPA | $10^{-9}$ M | 51 |
| Magnetic-bead biosensing platform | BPA | $4.3 \times 10^{-10}$ M | 52 |
| MN-PCDP | Anthracene | $10^{-9}$ M | This work |
| $Fe_3O_4@Ag$ MNPs-thiol | Anthracene | $10^{-6}$ M | 53 |

[a]MNPs: magnetic nanoparticles; diquat and paraquat have the similar structure.

respectively. The Raman detectable reproducibility of TMTD by hydrophobic slippery SERS platform is shown in Supplementary Figs. 24−30. In Fig. 4c, the Raman signals of TMTD characteristic peaks are acquired with 100% detection probability in 0.5 pM, ~65% in 50 fM, and ~10% in 5 fM. In addition, the solution-based aggregation approach, a simplest and effective way in commercial detection platforms at present, is adopted to clarify the consistency of SERS signal. As shown in Fig. 4d and Supplementary Figs. 31, 32, the SERS signals display superior spectral reproducibility and uniformity with a 100% detection probability and RSD value of ~5%, even at TMTD concentration of $10^{-12}$ M. In Fig. 4e, f and Supplementary Fig. 33, using the current enrichment-typed sensing strategy, the LODs of fluorescence detections for the concentrated and enriched molecules of carbendazim and BPA are also enhanced by 2−3 orders of magnitude. In this study, the enrichment protocol based on the adsorption and desorption processes of MN-PCDP adsorbent may significantly increase the sensitivity of plasmonic sensors, compared with the LOD for the same molecules with different SERS and fluorescence detection protocols (Table 1)[31–33]. Thus, the current strategy has wider applicability for mass spectrometry, chromatography, and other detection protocols.

**Separation and selective enrichment in real-sample complex matrix via current strategy.** Based on the distinguishing and selective absorption capacity for different molecules (Fig. 2 and Supplementary Figs. 10, 11)[29,34], the mesoporous nanosponge is expected to be used in the separation of interested molecules from

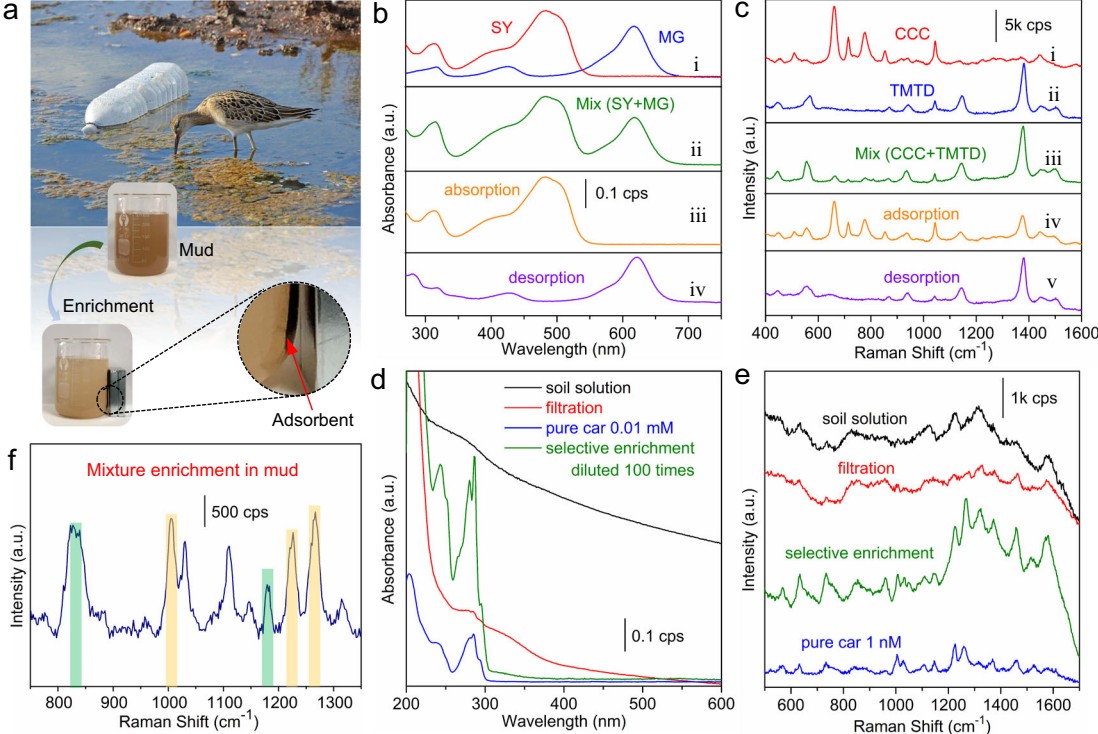

**Fig. 5 Separation and selective enrichment in complex matrix used this strategy. a** Optical photographs about the enrichment process of MN-PCDP in mud water. **b** UV−Vis spectra of SY (**i**, 0.01 mM), MG (**i**, 0.01 mM) and SY/MG (**ii**, both with 0.01 mM) mixture solution. **iii** The filtration mixture solution after the absorption of MN-PCDP for MG. **iv** The desorption solution of captured molecules (MG) redispersed in ethanol from the cavity of MN-PCDP adsorbent. **c** Raman spectra of CCC (**i**, 0.1 mM), TMTD (**ii**, 1 μM) and **iii** CCC (0.1 mM) /TMTD (1 μM) mixture solution before and after **iv** absorption, **v** desorption. **d** UV−Vis spectra about selective enrichment of organic pollutant molecules (carbendazim, 0.01 mM) from practical samples. **e** Raman spectra about selective enrichment of organic pollutant molecules (carbendazim, 1 nM) from practical samples. **f** Raman spectrum of mixture (BPA and carbendazim) after the enrichment process of MN-PCDP in real samples. Shaded areas mean the characteristic peaks of BPA (green, 830 and 1179 cm$^{-1}$) and carbendazim (yellow, 1004, 1222, and 1263 cm$^{-1}$) molecules.

the mixed systems. As shown in Fig. 5b and Supplementary Fig. 34, the malachite green (MG) and sunset yellow (SY) molecules are firstly mixed together and became to be the mixture solution (MG + SY) (Fig. 5b-i-ii). Then, after adsorption and separation processes using MN-PCDP, clearly, there is only SY left in the mixture solution (Fig. 5b-iii). Similarly, with a desorption process, the MG molecule is also successfully separated (Fig. 5b-iv). Following, two different pesticide molecules, namely chlormequat chloride (CCC) and TMTD, are used to further evaluate the selective separation using SERS detection (Fig. 5c-i-iii). Obviously, once TMTD is captured and separated, the SERS signals of CCC in the mixture solution significantly increases (Fig. 5c-iv), indicating that most TMTD molecules have been adsorbed. Moreover, the single Raman (Fig. 5c-v) and UV−Vis spectra (Supplementary Fig. 35) from TMTD molecules can be observed, illustrating only TMTD is effectively selected and separated by MN-PCDP.

Another advantage of the current enrichment protocol is that the interference of complex matrix can be effectively eliminated in the detection of real-sample system[35–37]. After the adsorption process, the MN-PCDP adsorbent can be easily separated by a magnet from a complicated environment containing, e.g., mud and microorganism (Fig. 5a). Figure 5d exhibits the UV−Vis spectra of carbendazim at a concentration of 0.01 mM in the filtered soil solution. It can be observed that the characteristic absorption peaks of carbendazim are very weak even after filtration. However, using the current selective separation and enrichment protocol, the absorption peaks intensity of carbendazim (Fig. 5d), BPA and MG (Supplementary Fig. 36a, b) are

respectively increased 463, 550, and 516 times comparing with pure solution. Similarly, as shown in Fig. 5e, when the concentration of detection molecules goes down to 1 nM, the SERS signals of carbendazim molecule in practical soil solution samples even after filtration are almost unobservable, whereas they can be easily detected after selective enrichment by MN-PCDP. The intensities of Raman characteristic peak from BPA and MG (Supplementary Fig. 37a, b) in complex matrix are also greatly improved.

In addition, in the real environment, more than one molecule is always studied, thus it is very important to realize the simultaneous detection of multiple molecules, especially in the existence of matrix interference. Figure 5f reveals that both the characteristic peaks of BPA (830 and 1179 cm$^{-1}$) and carbendazim (1004, 1222, and 1263 cm$^{-1}$) evidently appear in the Raman spectra of mixture solution, e.g., 1 μM BPA and 10 nM carbendazim, indicating the great absorption and detection capability for different molecules at the same time. Furthermore, the MN-PCDP demonstrates a superior reusability. As shown in Supplementary Fig. 38, six consecutive BPA adsorption/desorption cycles are performed and the regenerated MN-PCDP exhibited almost inappreciable decrease (90.2−84.3%) in performance compared to the as-synthesized polymer.

## Discussion

In summary, we have developed a robust and efficient sensing strategy based on the MN-PCDP mesoporous nanoponges to capture and enrich organic pollutants from water. In this strategy, the MN-PCDP adsorbent exhibits excellent selective adsorption

and enrichment capacity for various kinds of pollutants, eliminating the interference of complex matrix in the real-sample environments. Meanwhile, in diverse detection protocols of organic pollutants, e.g., UV−Vis, Raman, and fluorescent, the current sensing strategy significantly may increase the sensitivity with 2−3 orders of magnitude. Moreover, using the immobilization of magnetic NPs, the adsorption, separation, and enrichment processes by MN-PCDP can be completed within 2−3 min. Therefore, the current robust sensing strategy with the ultra-rapid, selective, and highly efficient molecule enrichment capability is believed to be applicable to a wider range of sensing devices for a cost-effective, simple, fast, flexible and portable detection. In the future, single-particle-MN-PCDP combining with Au NPs (SERS substrate) could dramatically lower the detection limit and enables higher spatial and temporal resolution[38–42], thus build single NP sensor to improve detection sensitivity (Supplementary Fig. 39).

## Methods

**Preparation of magnetite NPs (Fe₃O₄).** The carboxyl-functionalized magnetite NPs ($Fe_3O_4$) with highly water-dispersibility were synthesized by a modified solvothermal reaction approach[22]. Typically, $FeCl_3 \cdot 6H_2O$ (1.08 g, 4.0 mmol) and tri-sodium citrate (0.20 g, 0.68 mmol) were dissolved in ethylene glycol (20 mL) with stirring at 500 rpm. Afterward, sodium acetate trihydrate (2.0 g, 15 mmol) was added and the mixture was stirred for 30 min. Then, the mixture was sealed in a Teflon-lined stainless-steel autoclave (50 mL). The autoclave was heated at 200 °C for 12 h, and then allowed to cool to room temperature. The black products were washed with ethanol and ultrapure water for several times. Finally, the carboxyl-functionalized magnetite NPs ($Fe_3O_4$) were separated by magnet, re-dispersed in ethanol, and dried in vacuum drying oven at 30 °C.

**Preparation of magnetic NPs immobilized porous β-CD polymer (MN-PCDP).** The MN-PCDP composites were then prepared by modification of nucleophilic aromatic substitution method of hydroxyl groups of β-CD[20]. A dried 100 mL Shrek reaction vial with a magnetic stir bar was charged with β-CD (0.82 g, 0.724 mmol), TFT (0.40 g, 1.03 mmol), and $K_2CO_3$ (1.28 g, 9.28 mmol) and dried $Fe_3O_4$ (0.041 g). The vial was flushed with $N_2$ gas for 10 min, then an anhydrous THF/DMF mixture (9:1 v/v, 40 mL) was added and the vial was purged with $N_2$ for an additional 5 min. After that, the $N_2$ inlet was removed. The mixture was stirred at 500 rpm and refluxed at 85 °C for 36 h under nitrogen protection. The brown suspension was cooled to room temperature and magnetically separated the supernatant by magnet. The precipitate was washed twice with an appropriate amount of distilled water, THF, ethanol, and $CH_2Cl_2$, respectively. The final precipitate was vacuum dried at 77 K in a liquid nitrogen bath for 24 h and then the magnetic NPs immobilized porous β-CD polymer (MN-PCDP) was obtained.

**Batch adsorption kinetic studies.** In studies, the dried polymer (MN-PCDP, 20 mg) was initially washed with $H_2O$ for 2 times and then separated by a magnet. Adsorption kinetic studies for different pollutants were performed in 30 mL scintillation vials with 20 mL organic pollutant solution and 20 mg adsorbent, at ambient temperature on a hot plate at 25 °C. Then the sample was shaken at 250 rpm until the adsorption equilibrium was reached. The mixture was immediately stirred and 1 mL aliquots of the suspension were taken at certain intervals via syringe and filtered immediately by a 0.22 μm PTFE membrane filter. The residual concentration of the pollutant in each sample was determined by UV–vis spectroscopy.

**Calculation of removal efficiency.** The removal efficiency of pollutant removal by the adsorbent was determined by the following equation:

$$\text{Removal efficiency}(\%) = \frac{C_0 - C_t}{C_0} \times 100 \qquad (1)$$

where $C_0$ and $C_t$ are the initial and residual concentration of pollutant in the stock solution and filtrate, respectively.

**Flow-through adsorption experiments.** Individual pollutants were at high concentrations (mM). 5.0 mg of the MN-PCDP adsorbent was washed with deionized $H_2O$ for 2 times, then the precipitate was pushed by a syringe through a 0.22 μm PTFE membrane filter to form a thin layer of the adsorbent on the filter membrane. 5 mL of the pollutant stock solution was then pushed through the adsorbent in ~30 s (10 mL min⁻¹ flow rate). The filtrate was then measured by UV–Vis spectroscopy to determine the pollutant removal efficiency.

**MN-PCDP desorption studies.** 100.0 mg of the adsorbent was washed with deionized $H_2O$ for 2 times, and then added to the organic pollutant stock solution (0.01 mM) with determined volume (100, 250, and 500 mL). The mixture was shaken at 250 rpm for 1 min at 25 °C. After separating the supernatant and the adsorbent by an external magnet, the supernatant was filtered through a 0.22 μm filter membrane and determined by UV−Vis spectroscopy. Meanwhile, the precipitate was evaporated to dryness with a gentle nitrogen stream, then the residue was dissolved in 1 mL of ethanol to desorb the adsorbed organic pollutant. The desorption solution was measured by UV−Vis spectroscopy and compared with the initial concentration of pollutant in the stock solution.

**Calculation of enrichment efficiency.** The enrichment efficiency of pollutant adsorbed by the adsorbent was determined by the following equation:

$$\text{Enrichment efficiency} = \frac{C}{C_0}. \qquad (2)$$

Where $C_0$ and $C$ are the initial and desorbed solution concentration of pollutant, respectively.

**Fluorescence measurement.** The fluorescence spectra of pure solution were directly measured by a fluorescence spectrophotometer.

**Preparation of SERS active Au NPs.** The Au NPs with different sizes in diameter were synthesized based on a modified citrate reduction approach. The growth process of gold NPs with different sizes included three steps. For step 1, 100 mL of ultrapure water was added into a conical flask and heated to boiling. Then, 4 ml of 1 wt% sodium citrate (SC) solution was injected immediately, and 3.2 mL of 10 mM $HAuCl_4$ was added after 3 min. Kept the reaction for 25 min and made it natural cooling, then the Au seeds were obtained. For step 2, 80 mL of ultrapure water and 20 mL of Au seeds were mixed into the conical flask and heated to boiling. Then, 2 mL of 1 wt% sodium citrate solution was injected immediately, and 0.2 mL of $HAuCl_4$ was added 3 min later. Then additional 0.2 mL × 9 dosage of $HAuCl_4$ was injected every 8 min. After the last precursor was added, the reaction was kept for 25 min, and Au NPs-25 nm were obtained. For step 3, Au NPs prepared in step 2 were used as the seed solution, and the growth process was repeated as growth steps 2, and then Au NPs-55 nm were obtained in this step.

**SERS measurement.** SERS measurement is based on the hydrophobic slippery surface[17]. Concentrated molecules and Au NPs were prepared on a hydrophobic slippery Teflon membrane as follows: First, a Teflon membrane was attached on a flat glass slide (5 cm × 5 cm) by using a double-sided adhesive. Then, 0.5 mL of perfluorinated fluid was dispersed by spin coating. The low speed was 300 rpm for 30 s, and the high speed was 1500 rpm for 1 min. After the excess lubricating liquid was removed by centrifugal force, and the infused membrane was heated for 30 min. Lastly, 50 μL of probe molecules and 10 μL of Au colloids were simultaneously dropped onto the slippery surface. During drying, the contact line shrunk because of the low friction of the lubricated Teflon surface. As a result, the initial droplet could be concentrated into a small area less than 0.5 mm in diameter.

## Data availability

The data that support the findings of this study are available within the paper and its Supplementary Information or from the corresponding authors on reasonable request.

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

## Acknowledgements

This work was supported by the programs supported by the National Natural Science Foundation of China (No. 21675122, 21874104, 22074115 awarded to J.X.F. and 62022001 awarded to D.Y.L.), the Key Research Program in Shaanxi (2017NY-114 awarded to J.X.F.), and Natural Science Foundation of Shaanxi Province (No. 2019JLP-19 awarded to H.J.Y.), the World-Class Universities (Disciplines, awarded to J.X.F.) and the Characteristic Development Guidance Funds for the Central Universities (awarded to J.X.F.). The characterizations of materials are supported by the Instrument Analysis Center of Xi'an Jiaotong University.

## Author contributions

L.L.Z. synthesized the materials, carried out the characterizations and performance, analyzed the data, and wrote the draft of the manuscript. Y.G., R.H., Y.F.S., H.N., Y.Z.D., D.J.L. and D.Y.L. contributed in part of the TEM, Raman, and fluorescence characterizations. J.X.F. supervised the project, designed the experiments, contributed in discussions, comments, and writing of the manuscript. H.J.Y. designed the partial experiments, contributed in discussions and comments. All authors discussed the results and commented on the manuscript.

## Competing interests

The authors declare no competing of interest.
