## [Peer Review File · Nature Communications]

Reviewers' comments:

Reviewer #1 (Remarks to the Author):

The manuscript titled 'Ultra-rapid and highly efficient enrichment of organic pollutants via magnetic nanoparticles/mesoporous nanosponge compounds for ultrasensitive nanosensors' written by Zhang et al presented a strategy for enrichment sensing of organic pollutants based on a powerful porous composite material, consisting of magnetic nanoparticles immobilized porous β -cyclodextrin polymers. The results showed that the $\sim 90\%$ removal efficiency can reach within ~ 1 min and the polymer adsorbent can be easily recycled from water and re-dispersed in ethanol so that the target molecules in the cavity of adsorbent is concentrated, with an enrichment factor up to ~ 103 . The manuscript is carefully written and the logic is clear, so I think it belongs to Nature Communication with the following issues being addressed:

1. In the introduction section, the author introduced the background of surface-enhanced Raman scattering (SERS) and its limitations in detecting organic pollutants. I think this work has mainly reported a sample preparation method and doesn't face to resolve the scientific problems about SERS, so it is better to add some current sample preparation methods for organic pollutants detection based on magnetic nanoparticle and reduce some description about SERS limitations.
2. In Figure 2d, the FT-IR spectrum of the MN-PCDP displayed a new peak at 1265 cm^{-1} in relation to the newly formed C-F group, it is better to draw the chemical structure of the formed C-F bond. And I wonder the reasons about the selection of cross-linking agent TFT, and does the author has another try to combine the cross-linking agent and the β -CD? Figure 2a also showed the TEM images of MN, PCDP, and MN-PCDP. The high surface area and permanent porosity of MN-PCDP mesoporous nanosponge enable the rapid removals. It is better to show some screening experiments results in the supporting information to prove the optimal experimental condition.
3. In Figure 3a, the time-dependent adsorptions of various organic micropollutants adsorbed by MN-PCDP usually contain aromatic model compounds, does the absorption for organic molecules which are rich in alkane chain also work? And can the author also explain the mechanism about the high removal efficiency?
4. In Supplementary Figure 13, the fluorescence spectra of BPA before and after the enrichment of MN-PCDP adsorbent seems different and there are many spikes in Figure S13a and it is very smooth in Figure S13b, does it completely caused by the counts difference or exist other reasons? The author may unify it in both spectrums.
5. The study of nanoparticles sensing has been reached to single-particle level, such as single-nanoparticle electrochemistry. Also, some other nano-techniques for example, nanopore sensing based on single-molecule detection at confined space has been developed to improve the limits of detection (LOD). I wonder could the authors give some perspective on measurements science at single nanoparticle level in the manuscript. Please see the examples, ACS Sens. 2019, 4, 1185-1189., Anal. Chem. 2020, 92, 8, 5621-5644., Small Methods 2018, 1700390., Angew. Chem. Int. Ed. 2013, 52, 6011 -6014., Electrochem. Comm. ,2011, 13, 335-337.

Reviewer #2 (Remarks to the Author):

The manuscript is about the fabrication of a polymer based on the combination of beta-CD and magnetic NPs with high performance for rapid and highly efficient enrichment of organic pollutants. I found in the literature that the similar porous materials also combining magnetic NPs and CD have been already used for extraction of different organic molecules. Therefore the novelty of the paper is compromised.

Journal of Chromatography A, 1503 (2017) 1–111; <https://doi.org/10.1016/j.scitotenv.2020.138789>

Therefore the novelty is limited to the performance of the materials. Nevertheless, it is not demonstrated the selectivity or the performance in samples with mixture of pollutants. Therefore, this work is not suitable for Nat. Comm.

On the other hand the work is well presented and the data analysis is correct.

Reviewer #3 (Remarks to the Author):

My major concern is this study's novelty. Above of all, the concept of target enrichment is not novel, and magnetic nanoparticles functionalized with β -cyclodextrin have been extensively studied with widespread applications including organic pollutant sensing. In addition, the proposed method failed to offer a significantly improvement with respect to sensitivity and reproducibility compared to other related methods. Thus, it is not recommended for publication in Nature Communications.

Response to Reviewers' comments:

Reviewer #1 (Remarks to the Author):

The manuscript titled 'Ultra-rapid and highly efficient enrichment of organic pollutants via magnetic nanoparticles/mesoporous nanosponge compounds for ultrasensitive nanosensors' written by Zhang et al presented a strategy for enrichment sensing of organic pollutants based on a powerful porous composite material, consisting of magnetic nanoparticles immobilized porous β -cyclodextrin polymers. The results showed that the ~90% removal efficiency can reach within ~1 min and the polymer adsorbent can be easily recycled from water and re-dispersed in ethanol so that the target molecules in the cavity of adsorbent is concentrated, with an enrichment factor up to ~103. The manuscript is carefully written and the logic is clear, so I think it belongs to Nature Communication with the following issues being addressed:

1. In the introduction section, the author introduced the background of surface-enhanced Raman scattering (SERS) and its limitations in detecting organic pollutants. I think this work has mainly reported a sample preparation method and doesn't face to resolve the scientific problems about SERS, so it is better to add some current sample preparation methods for organic pollutants detection based on magnetic nanoparticle and reduce some description about SERS limitations.

Response: Thanks for the suggestion from the reviewer. With the enlightenment from such questions, we have rewrote the manuscript. In fact, it seems surprising that after fifty decades, SERS has not yet been widely used in practical applications. This is owing to the fact that, besides the stability of SERS substrates and reproducibility of spot-to-spot, SERS still faces two major bottlenecks in commercial market. The first is the low detection sensitivity to the molecules of intrinsic small cross-sections or weak affinity to metal surface. The second is the interference from the complex matrices in real-sample detection. With the revised version, we reconsidered the work and highlighted the capability of current detection strategy in selective adsorption to interested molecules, particularly from the complex matrix in the real-sample environment. This point is the key scientific problems about SERS. Certainly, the current enrichment strategy can be used in a wide detection protocols such as SERS, fluorescence, UV-vis, even Mass Spectrometry, Chromatography, and others.

Thus, in the revised version, we added some experiments and discussions about the SERS detection in real-sample environments, inspired by the comments from reviewers. The correlated

revised parts in the **Introduction section 2 and 3** has been updated.

2. In Figure 2d, the FT-IR spectrum of the MN-PCDP displayed a new peak at 1265 cm^{-1} in relation to the newly formed C-F group, it is better to draw the chemical structure of the formed C-F bond. And I wonder the reasons about the selection of cross-linking agent TFT, and does the author has another try to combine the cross-linking agent and the β -CD? Figure 2a also showed the TEM images of MN, PCDP, and MN-PCDP. The high surface area and permanent porosity of MN-PCDP mesoporous nanosponge enable the rapid removals. It is better to show some screening experiments results in the supporting information to prove the optimal experimental condition.

Response: We appreciate the suggestion from the reviewer. First, we are regretful for a mistake about the description of FT-IR peak at 1265 cm^{-1} . This peak in relation to the C-F group was existing in TFT, but the signal intensity of this absorption peak at 1265 cm^{-1} is weaker than that in TFT owing to the partial replacement of F, implying that the β -CD has been crosslinked with TFT. Second, as a comparison, in the revised version, we chose another cross-linking agent (epichlorohydrin, EPI), which is the most extensively studied β -CD polymer for water purification, to combine the β -CD. As shown in Supplementary Fig. 6, the removal efficiency of BPA by MNEPI-CDP was much lower than MNP-CDP. The correlated revised parts of the manuscript are shown as following:

Page 5, Paragraph 1: The signal intensity of absorption peak at 1265 cm^{-1} in relation to C-F stretching vibration is weaker than that in TFT owing to the partial replacement of F,^{23,24} implying that the β -CD has been crosslinked with TFT (Supplementary Fig.2).

Page 6, Paragraph 1: In this work, different cross-linking agent, e.g. epichlorohydrin (EPI) is compared (Supplementary Fig. 7). As shown in Supplementary Fig. 8, the removal efficiency of BPA by MNEPI-CDP in 1 min is 19.5%, which is much lower than MN-PCDP.

Supplementary Figure 2 | schematic about the synthesis of β -CD polymer (PCDP).

Supplementary Figure 7 | schematic about the synthesis of β -CD polymer crosslinked by EPI (EPI-CDP).

Supplementary Figure 8 | Uptake of pollutant by MNEPI-CDP. UV-vis spectra and removal efficiency recorded at different contact times of bisphenol A solution (0.1 mM) by MNEPI-CDP (1 mg mL^{-1}).

3. In Figure 3a, the time-dependent adsorptions of various organic micropollutants adsorbed by MN-PCDP usually contain aromatic model compounds, does the adsorption for organic molecules which are rich in alkane chain also work? And can the author also explain the mechanism about the high removal efficiency?

Response: Thanks for the advice from the reviewer. We added some absorption experiments of organic molecules with alkane chain to explain the mechanism about the high removal efficiency. The correlated revised parts of the manuscript are shown as following:

Page 6, Paragraph 2: As is known, the hydroxyl groups of β -CD are located at the outer surface of the molecule, that is, primary hydroxyls at the narrow side and secondary hydroxyls at the wider side, which makes β -CD water-soluble but simultaneously generates an inner cavity that is relatively hydrophobic.²⁷ Because of their hydrophobic interior cavity, β -CD can either partially or entirely accommodate suitably sized lipophilic low molecular weight molecules or even polymers.²⁸ For example, MN-PCDP exhibits a remarkable adsorption capability and selectivity

for most aromatics and some chain compounds, as shown in Fig. 2a and Supplementary Fig. 10-11. Furthermore, by means of particular treatments such as changing pH value of solution,²⁹ the adsorption feature of molecules can be tuned. Thus, the MN-PCDP mesoporous nanosponge will display a wide applicability and selectivity in a variety of molecules.

Supplementary Figure 10 | Uptake of pollutant by MN-PCDP. UV-vis spectra at different contact times of DMF (0.1 mM, a), PFOA (1 mM, b), CTAB (0.1 mM, c), CCC (1 mM, d), TMTD (0.1 mM, e) and HMTA (1 mM, f) with the adsorbent (1 mg mL⁻¹).

Supplementary Figure 11 | Uptake of pollutant by MNP-CDP. UV-vis spectra at different contact times of MG (0.01 mM, **a**) and SY (0.01 mM, **b**) with the adsorbent (1 mg mL^{-1}).

4. In Supplementary Figure 13, the fluorescence spectra of BPA before and after the enrichment of MN-PCDP adsorbent seems different and there are many spikes in Figure S13a and it is very smooth in Figure S13b, does it completely caused by the counts difference or exist other reasons? The author may unify it in both spectrums.

Response: Thanks for your reminder. In the Supplementary Figure 13, the concentration of BPA in the fluorescence spectra after the enrichment of MN-PCDP adsorbent was the original concentration of BPA. In this circumstance, after the enrichment of MN-PCDP adsorbent, the “concentration” of BPA was much high than the original. Thus, the Figure S13b seemed like more smooth. Here, in order to avoid this misunderstand, we used the lower concentration of BPA in Figure S13b and estimated the concentration after enrichment (according the enrichment factor ~ 500 fold). The correlated revised parts of the manuscript are shown as following:

Supplementary Figure 18 | Fluorescence spectra of BPA **a** before and **b** after the enrichment of MN-PCDP adsorbent. The concentration in the parentheses was estimated the concentration after enrichment (enrichment ~ 500 fold).

5. The study of nanoparticles sensing has been reached to single-particle level, such as single-nanoparticle electrochemistry. Also, some other nano-techniques for example, nanopore sensing based on single-molecule detection at confined space has been developed to improve the limits of detection (LOD). I wonder could the authors give some perspective on measurements science at single nanoparticle level in the manuscript. Please see the examples, ACS Sens. 2019, 4, 1185-

1189., *Anal. Chem.* 2020, 92, 8, 5621-5644., *Small Methods* 2018, 1700390., *Angew. Chem. Int. Ed.* 2013, 52, 6011–6014., *Electrochem. Comm.*, 2011, 13, 335–337.

Response: Thanks for your advice. Here, we give two kind of perspective about the single-molecule detection at confined space. The correlated revised parts of the manuscript are shown as following:

Page 11, Paragraph 1: In the future, single particle-MN-PCDP combining with Au NPs (SERS substrate) could dramatically lower the detection limit and enables higher spatial and temporal resolution,³⁵⁻³⁹ thus build single NP sensor to improve detection sensitivity (Supplementary Fig. 24).

Supplementary Figure 24 | Schematic of measurements at single nanoparticle level. by MN-PCDP. **a** Single Fe₃O₄-Au core-shell structure nanoparticle with crosslinked porous β-CD polymer on the surface, detection solution with target molecular is concentrated from 50 μL to 3.4×10⁻⁸ μL (enrichment ~1.5×10⁹ fold). **b** MN-PCDP with Au NPs on the surface, detection solution with target molecular is concentrated from 50 μL to 4.2×10⁻⁶ μL (enrichment ~1.2×10⁷ fold).

References

- 35 Li, Q. et al. Detection of single proteins with a general nanopore sensor. *ACS Sens.* **4**, 1185–1189, doi: 10.1021/acssensors.9b00228 (2019).
- 36 Lu, S. et al. Electrochemical sensing at a confined space. *Anal. Chem.* **92**, 5621–5644, doi: 10.1021/acs.analchem.0c00931 (2020).

- 37 Ying, Y. et al. Electrochemical confinement effects for innovating new nanopore sensing mechanisms. *Small Methods* **2**, 1700390, doi: 10.1002/smt.201700390 (2018).
- 38 Shi, L. et al. Plasmon resonance scattering spectroscopy at the single nanoparticle level: real-time monitoring of a click reaction. *Angew. Chem. Int. Ed.* **52**, 6011–6014, doi: 10.1002/ange.201301930 (2013).
- 39 Gao, Q. et al. Highly sensitive impedimetric sensing of DNA hybridization based on the target DNA-induced displacement of gold nanoparticles attached to ssDNA probe. *Electrochem. Commun.* **13**, 335–337, doi: 10.1016/j.elecom.2011.01.018 (2013).

Reviewer #2 (Remarks to the Author):

The manuscript is about the fabrication of a polymer based on the combination of beta-CD and magnetic NPs with high performance for rapid and highly efficient enrichment of organic pollutants. I found in the literature that the similar porous materials also combining magnetic NPs and CD have been already used for extraction of different organic molecules. Therefore, the novelty of the paper is compromised. Journal of Chromatography A, 1503 (2017) 1–11; <https://doi.org/10.1016/j.scitotenv.2020.138789>. Therefore the novelty is limited to the performance of the materials. Nevertheless, it is not demonstrated the selectivity or the performance in samples with mixture of pollutants. Therefore, this work is not suitable for Nat. Comm.

Response: Thanks for your comments. After receiving the comments from referees, we noticed that probably, referees provided such comments is because that we cannot demonstrate the novelty and significance of this work clearly in the introduction part. In fact, after carefully read the suggested two papers, i.e. *Science of the Total Environment* 728 (2020) 138789 and *Journal of Chromatography A, 1503 (2017) 1–11*, we would like to highlight the concept of extraction in mentioned two papers and our “enrichment” in our manuscript. In addition, with the enlightenment from referees, we have noticed that the selectivity and the performance in detection of complex matrix in real-sample environment indeed are very important. Thus, in the revised version, we have updated these data, and rewrote the manuscript. In the following, we shall explain in details.

1) Concerning the concept of extraction in mentioned two papers and our “enrichment” in our

manuscript. I have read the extraction related references (including the suggested papers, Science of the Total Environment 728 (2020) 138789 and Journal of Chromatography A, 1503 (2017) 1–11). Yes, they indeed used the similar porous materials. However, reviewer is interesting of “the selectivity or the performance of the materials” since extraction is to select target molecule from complex system. E.g., extraction (adsorption) of organic molecules (microcystins or PAHs) from environmental water or soil sample in suggested two papers. **Selectivity** is the first importance in this kind of work, i.e. selective adsorption interested molecules from multiple-molecules system. **They did not notice to improve the removal speed, and enrichment capability**, thus you may see that in above two papers, **one used nearly one hour to extract and 7 min to desorb and the other paper used high concentration adsorption materials and desorbed in a large volume of desorption solvent**. The extraction process if operation into a traditional solid state extraction cartridges, **the separating rate is relatively slow**. Normally, the speed of extraction is 0.5-5 mL/min, which can not meet our current study motivation with very fast enrichment operation. Extraction seems to be more the work for environmental scientists, e.g. water or soil treatment.

However, in our current work, we are aiming our work to develop a new detection strategy for trace pollutants, and we paid our attention to the enrichment capability and fast operation, (in the following, we shall list the novelty and advance of this work), which is very important for the practical application in portable, fast, on site detection. Thus, our enrichment strategy is different from classical extraction process.

With the enlightenment of referees, we reconsidered and updated the significance and novelty of this work, and we want to emphasize here: we explored a very effective enrichment and detection strategy with remarkable performance including

- 1) ultra-fast removal speed within ten second to ~1 min,
- 2) high removal efficiency up to ~90%,
- 3) ultra-fast desorption speed, within ten seconds to ~1 min,
- 4) high desorption efficiency up to ~100%,

5) selectivity to interested target molecules thus benefit to the detection of real-samples free from interference of impurities or fluorescence background (this point is new enlightened by referees),

- 6) totally high enrichment factor thus high increased sensitivity up to 10(3) level within 2~3

min operation period which is very critical for the practical application in portable, fast, on site detection.

7) wide application potential for SERS, fluorescence, UV-vis and other sensing strategies.

In addition, to improve our enrichment capability and fast operation, we referred a recent reported best processes to prepare the mesoporous -cyclodextrin polymer published in Nature, 2016, 529, 190, cited in our manuscript in ref 20 in the revised version. We added magnetic particles into this best mesoporous CD polymer. If you compare our results, you may found we demonstrated the best and carefully designed protocols so that we realized (repeated here again) **above seven performance.**

In the revised version, with the enlightenment from referees, we have updated the data and added the new experiments on the selectivity and the performance in detection of complex matrix in real-sample environment, which are indeed very important.

In our revised version of manuscript, we have added Figure 5, and related text description, and also some data in supporting information. These include,

Fig. 5a shows that the adsorbent was easily collected on the wall of beaker with a magnet by our porous composite material from the **complex matrix**, such as mud and microorganism.

Page 9, Paragraph 2: Based on the distinguishing and selective absorption capacity for different molecules (Fig. 2 and Supplementary Fig. 10-11),³⁴ the mesoporous nanosponge is expected to be used in the separation of interested molecules from the mixed systems. As shown in Fig. 5b and Supplementary Fig. 19a-c, the MG (malachite green) and sunset yellow (SY) molecules are firstly mixed together and became to be the mixture solution (MG+SY) (Fig. 5b-i-ii). Then, after adsorption and separation processes using MN-PCDP, clearly, there is only SY left in the mixture solution (Fig. 5b-iii). Similarly, with a desorption process, the MG molecule is also successfully separated (Fig. 5b-iv). Following, two different pesticide molecules, namely CCC and TMTD, are used to further evaluate the selective separation using SERS detection (Fig. 5c-i-iii). Obviously, once TMTD is captured and separated by MN-PCDP adsorbent, the SERS signals of CCC in the mixture solution significantly increases (Fig. 5c-iv), indicating that most TMTD molecules have been adsorbed. Moreover, only TMTD molecules from Raman (Fig. 5c-v) and UV-vis spectra (Supplementary Fig. 20) can be observed, illustrating only TMTD is effectively selected and separated by MN-PCDP.

Page 9, Paragraph 3: Another advantage of the current enrichment protocol is that the

interference of complex matrix can be effectively eliminated in the detection of real-sample system. After adsorption process, the MN-PCDP adsorbent can be easily separated by a magnet from a complicated environment containing, e.g. mud and microorganism (Fig. 5a). Fig. 5d and Supplementary Fig. 21 exhibit the UV-vis spectra of carbendazim at a concentration of 0.01 mM in the filtered soil solution. It can be observed that the characteristic absorption peaks of carbendazim are very weak even after filtration. However, using current selective separation and enrichment protocol, the absorption peaks intensity of carbendazim (Fig. 5d), BPA (Supplementary Fig. 21a) and MG (Supplementary Fig. 21b) are respectively increased 463, 550, and 516 times comparing with pure carbendazim solution. Similarly, as shown in Fig. 5e and Supplementary Fig. 22, when the concentration of detection molecules goes down to 0.1 μM , the SERS signals of carbendazim molecule in practical soil solution samples even after filtration are almost unobservable, whereas they can be easily detected after selective enrichment by MN-PCDP.

Page 10, Paragraph 2: In addition, in the real environment, more than one molecule is always studied, thus it is very important to realize the simultaneous detection of multiple molecules, especially in the existence of matrix interference. Fig. 5f reveals that both the characteristic peaks of BPA (830 and 1179 cm^{-1}) and carbendazim (1008, 1244 and 1263 cm^{-1}) evidently appear in the Raman spectra of mixture solution, e.g. 1 μM BPA and 10 nM carbendazim, indicating the great absorption and detection capability for different molecules at the same time

Fig. 5 Separation and selective enrichment in complex matrix used this strategy. a Optical photographs about the enrichment process of MN-PCDP in mud water. **b** UV-vis spectra of SY (i, 0.01 mM), MG (i, 0.01 mM) and SY/MG (ii, both with 0.01 mM) mixture solution before and after absorption, desorption. **iii** The filtration mixture solution after the absorption of MN-PCDP for MG. **iv** The desorption solution of captured molecules (MG) redispersed in ethanol from the cavity of MN-PCDP adsorbent. **c** Raman spectra of CCC (i, 0.1 mM), TMTD (ii, 1 μ M) and **iii** CCC (0.1 mM) /TMTD (1 μ M) mixture solution before and after **iv** absorption, **v** desorption. **d** UV-vis spectra about selective enrichment of organic pollutant molecules (carbendazim, 0.01 mM) from practical samples. **e** Raman spectra about selective enrichment of organic pollutant molecules (carbendazim, 0.1 μ M) from practical samples. **f** Raman spectrum of mixture after the enrichment process of MN-PCDP in real samples.

Supplementary Figure 19 | Separation of SY/MG mixture dyes: **a** photograph of SY (0.01 mM), MG (0.01 mM) and SY/MG (both with 0.01 mM) mixture solution, **b** photograph of SY/MG mixture solution after absorption by MN-PCDP, **c** photograph of SY/MG mixture solution after desorption by MN-PCDP.

Supplementary Figure 20 | Separation of CCC/TMTD mixture pesticides: UV-vis spectra of CCC (1 mM), TMTD (0.1 mM) and CCC (1 mM)/TMTD (0.1 mM) mixture solution before and after absorption, desorption.

Supplementary Figure 21 | UV-vis spectra about selective extraction and enrichment of organic pollutant molecules (0.01 mM) from practical samples: **a** BPA in industrial wastewater, **b** MG in pond water.

Supplementary Figure 22 | Raman spectra about selective extraction and enrichment of organic pollutant molecules (0.01 mM) from practical samples: **a** BPA in industrial wastewater, **b** MG in pond water.

Reviewer #3 (Remarks to the Author):

My major concern is this study's novelty. Above of all, the concept of target enrichment is not novel, and magnetic nanoparticles functionalized with β -cyclodextrin have been extensively studied with widespread applications including organic pollutant sensing. In addition, the proposed method failed to offer a significantly improvement with respect to sensitivity and reproducibility compared to other related methods. Thus, it is not recommended for publication in Nature Communications.

Response: Thanks for your comments. After receiving the comments from referees, we noticed that probably, referees provided such comments because that we cannot demonstrate the novelty and significance of this work clearly in the introduction part. In fact, after carefully investigate the comments, two points are including, one is that referees said the similar materials have been reported, and the other is that this kind materials has been used for sensing.

Concerning above two points, in the revised version, we have rewrote the introduction, and supplemented new experiments and added them.

For the materials, we want to explain two concept of “extraction” in some publications and our “enrichment” in our manuscript. I have read many extraction related references. Yes, they indeed used the similar porous materials. However, these materials in these publications were mainly used for extraction, which is to select target molecule from complex system. E.g., extraction (adsorption) of organic molecules (microcystins or PAHs) from environmental water or soil sample in suggested two papers. **Selectivity** is the first importance in this kind of work, i.e. selective adsorption interested molecules from multiple-molecules system. **They did not notice to improve the removal speed, and enrichment capability**, thus you may see that in above two papers, **one used nearly one hour to extract and 7 min to desorb and the other paper used high concentration adsorption materials and desorbed in a large volume of desorption solvent**. The extraction process if operation into a traditional solid state extraction cartridges, **the separating rate is relatively slow**. Normally, the speed of extraction is 0.5-5 mL/min, which can not meet our current study motivation with very fast enrichment operation. Extraction seems to be more the work for environmental scientists, e.g. water or soil treatment.

However, in our current work, we are aiming our work to develop a new detection strategy for trace pollutants, and we paid our attention to the enrichment capability and fast operation, (in the following, we shall list the novelty and advance of this work), which is very important for the practical application in portable, fast, on site detection. Thus, our enrichment strategy is different from classical extraction process.

With the enlightenment of referees, we reconsidered and updated the significance and novelty of this work, and we want to emphasize here: we explored a very effective enrichment and detection strategy with remarkable performance including

- 1) ultra-fast removal speed within ten second to ~1 min,

- 2) high removal efficiency up to ~90%,
- 3) ultra-fast desorption speed, within ten seconds to ~1 min,
- 4) high desorption efficiency up to ~100%,

5) selectivity to interested target molecules thus benefit to the detection of real-samples free from interference of impurities or fluorescence background (this point is new enlightened by referees).

6) totally high enrichment factor thus a high increased sensitivity up to 10³ level within 2~3 min operation period which is very critical for the practical application in portable, fast, on site detection.

- 7) wide application potential for SERS, fluorescence, UV-vis and other sensing strategies.

In addition, to improve our enrichment capability and fast operation, we referred a recent reported best processes to prepare the mesoporous β -cyclodextrin polymer published in Nature, 2016, 529, 190, cited in our manuscript in ref 20 in the revised version. We added magnetic particles into this best mesoporous CD polymer. If you compare our results, you may found we demonstrated the best and carefully designed protocols so that we realized (repeated here again) above seven performance.

Concerning the comments of referee “this kind materials has been used for sensing”, we have carefully studied the literatures and according to our knowledge, the organic pollutant **sensing** from literatures based on the magnetic nanoparticles functionalized with β -cyclodextrin, can be divided into **two categories**. **One** is that some magnetic materials decorated with β -cyclodextrin **monomer** to build up such as electrochemical sensing. In this case, they did not use crosslinked β -cyclodextrin polymers and the absorption capacity of β -cyclodextrin monomer was much less than the crosslinked β -cyclodextrin polymers. Moreover, they did not carry out enrichment related study. **The other kind of** sensing using this kind of materials is to use crosslinked β -cyclodextrin polymers, but they use for liquid chromatography (HPLC) or photoluminescence spectroscopy sensing, e.g. solid-phase extraction adsorbent. Again, they are studying the extraction and selectivity of target molecule from complex pollutants.

Above two kinds of sensing are quite different as our enrichment and concentration-strategy for detection. More importantly, up to now, we have not found the reports of SERS or fluorescence related sensing based on the current reported highly effective enrichment strategy of magnetic

nanoparticles immobilized with crosslinked porous β -CD polymer.

Actually, like the nature paper we cited in Ref. 20 in the revised version, the crosslinked porous β -CD polymer have also reported before this work, but this work again reported an improved and excellent performance. Thus we think it is valuable to be published again in Nature. For our work, besides of above seven remarkable performance. We cited this excellent method in Nature, and we added new magnetic design and obtain excellent enrichment properties, and selectivity capability to eliminate the interference of complex matrix in the real-sample detection environment.

Therefore, the novelty and significance of our manuscript are still there and we are believing it is valuable to be considered for publish in Nature Communications.

As for “the proposed method failed to offer a significantly **improvement with respect to sensitivity and reproducibility** compared to other related methods.” We listed seven remarkable performance of the current strategy. Actually, item-2), 4), and 6) are related to this questions. Using the current protocol, we realized high efficient adsorption up to ~90%, selective separation by means of the effects of host-guest inclusion and magnet, desorption with nearly ~100% efficiency, thus the total enrichment capability up to ~1000 times, hence an improved and increased sensitivity up to 10(3) level can be obtained. Particularly, within 2~3 min operation period which is very critical for the practical application in portable, fast, on site detection. These data have been shown in Figure 4 and Figure 5 according to the detection method of SERS or fluorescence spectroscopies.

As for the reproducibility, in the Fig. S22, we provided the adsorption efficiency of different experiment times and consecutive regeneration cycles, indicating the great reproducibility.

In addition, with the enlightenment from referees, we have added many new data in the revised version and we rewrote the manuscript, we added the new results on the selectivity to interested target molecules, thus benefit to the detection of real-samples free from interference of impurities or fluorescence background (Presented in new Figure 5 and supporting information 19-22).

Therefore, I sincerely hope this revised version with many supplemented data can be considered and accept for publication as soon as possible.

REVIEWER COMMENTS

Reviewer #1 (Remarks to the Author):

The authors addresses my comments very well. I learned a lot from it. Thanks you, I recommend the publication in Nature Communications.

Reviewer #2 (Remarks to the Author):

I think that the authors have made a lot of effort in improving the manuscript, they have included further data regarding the selectivity and multiplexing capabilities of the sensing and extraction platform. But from my point of view, the work is not relevant enough to publish in this journal. I think that it is more suitable for a sister journal.

Reviewer #3 (Remarks to the Author):

By introducing magnetic nanoparticles to an established system (Nature, 2016, 529, 190–194), this study achieved ultra-rapid and highly efficient enrichment of organic pollutants. Magnetic SERS has been extensively studied and applied in organic pollutant sensing (Journal of Environmental Sciences, 2019, 80, 14-34). Compared with those related researches, the proposed study mainly focused on removal speed, and enrichment capability. However, sensitivity and spectral reproducibility are two crucial factors for SERS-based nanosensors, which are more important and should not be neglected. The enrichment ability of such materials makes sense for me, but as an ultrasensitive nanosensor, the following issues haven't been addressed in the revised manuscript.

1. No significant improvement in sensitivity has been achieved in this study, comparing with other magnetic SERS-based sensors for organic pollutants.
2. The authors did nothing to improve the SERS spectral reproducibility, so the accuracy of the proposed method is not convincing.
3. In my opinion, a SERS nanosensor that combines ultrahigh sensitivity (achieved by high enrichment ability and signal enhancement ability), high spectral reliability, and fast removal speed, is publishable in the high-level journal of Nature Communication.

Response to Reviewer' comments

Reviewer #1 (Remarks to the Author):

The authors addresses my comments very well. I learned a lot from it. Thanks you, I recommend the publication in Nature Communications.

Response: Thank you very much for the reviewer's comments and suggestions.

Reviewer #2 (Remarks to the Author):

I think that the authors have made a lot of effort in improving the manuscript, they have included further data regarding the selectivity and multiplexing capabilities of the sensing and extraction platform. But from my point of view, the work is not relevant enough to publish in this journal. I think that it is more suitable for a sister journal.

Response: Thank you very much for the reviewer's comments, just as the recommendation from reviewer-1 and -3, we believe a highly efficient and ultrafast enrichment protocol and a high sensitivity, fast operation, and good repeatability of SERS detection is important to the field and valuable to be published in Nature Communications.

Reviewer #3 (Remarks to the Author):

By introducing magnetic nanoparticles to an established system (Nature, 2016, 529, 190–194), this study achieved ultra-rapid and highly efficient enrichment of organic pollutants. Magnetic SERS has been extensively studied and applied in organic pollutant sensing (Journal of Environmental Sciences, 2019, 80, 14-34). Compared with those related researches, the proposed study mainly focused on removal speed, and enrichment capability. However, sensitivity and spectral reproducibility are two crucial factors for SERS-based nanosensors, which are more important and should not be neglected.

The enrichment ability of such materials makes sense for me, but as an ultrasensitive nanosensor, the following issues haven't been addressed in the revised manuscript.

1. No significant improvement in sensitivity has been achieved in this study, comparing with other magnetic SERS-based sensors for organic pollutants.

Answer: Thanks for the advice from the reviewer. In this manuscript, we realize highly efficient

enrichment capability up to ~1000 times and the LODs of fluorescence detections for enriched molecules were enhanced by 2~3 orders. Depending on the reviewer comment, in the revised manuscript, the SERS sensitivities of our enrichment method were measured and compared with other reports. As for the SERS sensitivity for the organic pollutants, we chose a more common molecular, TMTD (also called thiram), as the probe molecule to explain the superiorities of our protocol. As shown in Fig. 4a-b, the LOD of TMTD after the enrichment of MN-PCDP is up to 5 fM (Fig. 4c), that is superior to most of the magnetic SERS-based sensors (generally no better than 10^{-12} M). Moreover, as for carbendazim and diquat, the detectabilities are obtained in 5 pM (Supplementary Fig. 18) and 1 pM (Supplementary Fig. 19), respectively, while the similar SERS sensor just achieve to ~nM level. Furthermore, the detectabilities of several analytes are summarized in Table 1, comparing with the similar magnetic SERS-based sensors, the LODs of TMTD, carbendazim, diquat, BPA and anthracene are reached in great sensitivity. The correlated revised revisions are shown as following:

Page 8, Paragraph 2: As shown in Figure 4a-b, without the enrichment process, the LOD of SERS for TMTD is around 1 pM. However, after the enrichment using MN-PCDP adsorbent, this value reaches to ~5 fM, showing an increase of $10^2\sim 10^3$. Meanwhile, based on this enrichment protocol, the detachabilities of BPA, carbendazim, diquat anthracene are up to 0.1 nM, 5 pM, 1 pM, and 1 nM (Supplementary Fig. 17-20), which are much lower than most of the magnetic SERS-based sensors (Table 1).

Fig. 4 Enhanced Raman spectra of TMTD **a** before and **b** after enrichment process of MN-PCDP.

Supplementary Figure 19 | Enhanced Raman spectra of diquat **a** before and **b** after the enrichment of MN-PCDP adsorbent. The characteristic peaks of diquat were at 1061, 1170, 1376 and 1563 cm^{-1} .

Supplementary Figure 20 | Enhanced Raman spectra of anthracene **a** before and **b** after the enrichment of MN-PCDP adsorbent. The characteristic peaks of anthracene were at 748, 1160, 1389, and 1540 cm^{-1} .

Table 1 Detectability of the MNPs for organic pollutants reported in the literature

Types of MNPs ^a	Analyte	LOD	Ref.
MN-PCDP	TMTD	5×10^{-15} M	This work
$\text{Fe}_3\text{O}_4@SiO_2@Ag$ nanospindles	TMTD	10^{-7} M	[31]
$\text{Fe}_3\text{O}_4@Au$ NRs array	TMTD	10^{-9} M	[32]
$\text{Fe}_3\text{O}_4@SiO_2@Ag$ core-shell MNPs	TMTD	10^{-9} M	[33]
Cube-like $\text{Fe}_3\text{O}_4@SiO_2@Au@Ag$	TMTD	5×10^{-11} M	[34]
Flower-like $\text{Fe}_3\text{O}_4@SiO_2@Ag$	TMTD	10^{-11} M	[35]
$\text{Fe}_3\text{O}_4@Ag-PEI-Au@Ag$	TMTD	5×10^{-12} M	[36]
$Au@Fe_3O_4$ network	TMTD	5×10^{-14} M	[37]

MN-PCDP	Diquat	10^{-12} M	This work
Fe ₃ O ₄ @AuNRs assemblies	Diquat	10^{-9} M	[32]
Fe ₃ O ₄ @Ag-PEI-Au@Ag	Paraquat ^b	10^{-10} M	[36]
MN-PCDP	Carbendazim	5×10^{-12} M	This work
Au/Fe ₃ O ₄ nanocomposite	Carbendazim	2.3×10^{-9} M	[38]
MN-PCDP	BPA	10^{-10} M	This work
Magnetic gold nanoclusters	BPA	10^{-9} M	[39]
Magnetic-bead biosensing platform	BPA	4.3×10^{-10} M	[40]
MN-PCDP	Anthracene	10^{-9} M	This work
Fe ₃ O ₄ @Ag MNPs-thiol	Anthracene	10^{-6} M	[41]

^aMNPs: magnetic nanoparticles; diquat and paraquat have the similar structure.

References

- 31 He, Q. *et al.* Fabrication of Fe₃O₄@SiO₂@Ag magnetic-plasmonic nanospindles as highly efficient SERS active substrates for label-free detection of pesticides. *New J. Chem.* **41**, 1582-1590, doi: 10.1039/c6nj03335k (2017).
- 32 Tang, S. *et al.* Efficient enrichment and self-assembly of hybrid nanoparticles into removable and magnetic SERS substrates for sensitive detection of environmental pollutants. *ACS Appl. Mater. Interfaces* **9**, 7472-7480, doi: 10.1021/acsami.6b16141 (2017).
- 33 Wang, C. *et al.* Seed-mediated synthesis of high-performance silver-coated magnetic nanoparticles and their use as effective SERS substrates. *Colloids and Surfaces A: Physicochem. Eng. Aspects* **506**, 393-401, doi: 10.1016/j.colsurfa.2016.05.103 (2016).
- 34 Sun, M. *et al.* Cube-like Fe₃O₄@SiO₂@Au@Ag magnetic nanoparticles: a highly efficient SERS substrate for pesticide detection. *Nanotechnology* **29**, 165302, doi: 10.1088/1361-6528/aaae42 (2018).
- 35 Wang, C. *et al.* Sonochemical synthesis of highly branched flower-like Fe₃O₄@SiO₂@Ag microcomposites and their application as versatile SERS substrates. *Nanoscale* **8**, 19816-19828, doi: 10.1039/c6nr07295j (2016).

- 36 Wang, C. *et al.* Polyethylenimine-interlayered core-shell-satellite 3D magnetic microspheres as versatile SERS substrates. *Nanoscale* **7**, 18694-18707, doi: 10.1039/c5nr04977f (2015).
- 37 Yang, T. *et al.* Au dotted magnetic network nanostructure and its application for on-site monitoring femtomolar level pesticide. *Small* **10**, 1325-1331, doi: 10.1002/sml.201302604 (2014).
- 38 Li, Q. *et al.* A gold/Fe₃O₄ nanocomposite for use in a surface plasmon resonance immunosensor for carbendazim. *Microchimica Acta* **186**, 313, doi: 10.1007/s00604-019-3402-0 (2019).
- 39 Kadasala, N. R. & Wei, A. Trace detection of tetrabromobisphenol A by SERS with DMAP-modified magnetic gold nanoclusters. *Nanoscale* **7**, 10931-10935, doi: 10.1039/c4nr07658c (2015).
- 40 Xiao, R., Wang, W. C., Zhu, A. N. & Long, F. Single functional magnetic-bead as universal biosensing platform for trace analyte detection using SERS-nanobioprobe. *Biosens. Bioelectron.* **79**, 661-668, doi: 10.1016/j.bios.2015.12.108 (2016).
- 41 Du, J. J. & Jing, C. Y. Preparation of thiol modified Fe₃O₄@Ag magnetic SERS probe for PAHs detection and identification. *J. Phys. Chem. C* **115**, 17829-17835, doi: 10.1021/jp203181c (2011).
- 46 Song, D. *et al.* A label-free SERRS-based nanosensor for ultrasensitive detection of mercury ions in drinking water and wastewater effluent. *Anal. Methods* **9**, 154–162, doi: 10.1039/c6ay02361d (2017).

2. The authors did nothing to improve the SERS spectral reproducibility, so the accuracy of the proposed method is not convincing.

Answer: Thanks for the suggestion from the reviewer. Following the suggestion, the reproducibility of the enrichment SERS detection was studied in the revised manuscript. In current protocol, the excellent SERS performance is achieved by the enrichment (adsorption and desorption) from the MN-PCDP adsorbent. For the SERS spectral reproducibility, **first of all**, we measured the removal efficiencies and enrichment efficiencies from different independent experiments. As shown in Supplementary Fig. 21-23 and Supplementary Table 5-6, the relative standard deviations (RSDs) of

the removal efficiencies and enrichment efficiencies for target molecules (BPA, carbendazim, TMTD diquat and anthracene) are less than 1% and 5%, respectively, indicating that the enrichment performance of MN-PCDP adsorbent is stable and effective. **Second**, we optimized the detection method based on the hydrophobic slippery SERS platform to further improve the SERS reproducibility. As shown in Supplementary Fig. 24, initial droplets of 50 μ L (contain Au NPs and analyte) are concentrated into smaller spots with diameter less than 0.25 mm. Meanwhile, the enhanced Raman spectra of TMTD in 1 pM, 0.5 pM, 0.1 pM, 50 fM, 10 fM and 5 fM are shown in Supplementary Fig. 25-30, indicating the good reproducibility of this magnetic SERS-based sensor. Another SERS substrate, the wet-chemical synthesized Au NPs with addition of inorganic salt, was adopted to explain the consistency of SERS signal, which was the simplest and effective way in commercial detection platforms at present. For the aggregating approach with Au colloid, the reproducibility was consistent in aggregation states from different batches. As shown in Fig 4d and Supplementary Fig. 31-32, the detection probability is up to 100%. **Third**, the SERS detection about analyte with lower concentration in real-sample system was also tested. In Fig. 5e, carbendazim molecule with 1 nM in soil solution is distinctly detected by the selective enrichment of MN-PCDP. The correlated revisions are shown as following:

Page 9, Paragraph 1: Furthermore, multiple adsorption and desorption experiments by MN-PCDP for above five organic molecules are implemented to illustrate the reproducibility of this adsorbent. In Supplementary Fig. 21-23, Supplementary Table 5-6, the removal efficiencies and enrichment efficiencies of MN-PCDP adsorbent are excellent for target molecules with RSD less than 1% and 5%, respectively. The Raman detectable reproducibility of TMTD by hydrophobic slippery SERS platform is shown in Supplementary Fig. 24-30. In Fig. 4c, the Raman signals of TMTD characteristic peaks are acquired with 100% detection probability in 0.5 pM, ~65% in 50 fM and ~10% in 5 fM. In addition, the solution-based aggregation approach, a simplest and effective way in commercial detection platforms at present, is adopted to clarify the consistency of SERS signal. As shown in Fig 4d and Supplementary Fig. 31-32, the SERS signals display superior spectral reproducibility and uniformity with 100% detection probability and RSD value of ~5%, even at TMTD concentration of 10^{-12} M.

Fig. 4 c Probability of SERS signals with different concentrations of TMTD after enrichment process of MN-PCDP by hydrophobic slippery SERS platform. The inserted schematic diagram shows hydrophobic slippery SERS platform. **d** Detection probability (red) and intensity of SERS signals (black) with different concentrations of TMTD after enrichment process of MN-PCDP by aggregating approach with Au colloid. The inserted schematic diagram shows aggregating approach with Au colloid.

Supplementary Figure 21 | Adsorption reproducibility of MN-PCDP. UV-vis spectra recorded as a function of different absorption experiments by MN-PCDP for removal BPA (0.1 mM, **a**), carbendazim (0.1 mM, **b**), TMTD (0.1 mM, **c**), diquat (0.01 mM, **d**) and anthracene (0.01 mM, **e**) by MN-PCDP (1 mg mL⁻¹).

Supplementary Table 5 | The enrichment efficiency of BPA, carbendazim, TMTD, diquat and anthracene by different absorption cycles in Figure S21

Removal efficiency (%)	BPA	Carbendazim	TMTD	Diquat	Anthracene
1	89.17	91.37	69.20	96.17	93.24
2	90.01	92.67	68.98	96.05	94.59
3	89.59	91.62	69.29	96.03	94.59
4	88.63	90.94	69.17	96.42	93.24
5	89.01	90.94	69.15	96.37	93.24
6	87.51	92.31	68.42	94.92	93.24
7	89.71	91.20	70.15	94.70	94.59
8	90.51	91.21	69.74	95.06	93.24
9	88.26	90.78	69.41	96.08	93.24
10	88.93	91.92	69.92	95.89	94.59
Average	89.13	91.49	69.34	96.17	94.78
RSD	0.98%	0.69%	0.71%	0.66%	0.074%

Supplementary Figure 22 | Enrichment reproducibility of MN-PCDP. UV-vis spectra recorded as a function of triplicate desorption experiments by MN-PCDP for enrichment BPA (0.01 mM, **a**), carbendazim (0.01 mM, **b**), TMTD (0.01 mM, **c**), diquat (0.01 mM, **d**) and anthracene (0.01 mM, **e**) by MN-PCDP (100 mg in 250 mL×4 times absorption).

Supplementary Table 6 | The enrichment efficiency of BPA, carbendazim, TMTD, diquat and anthracene by different enrichment cycles in Figure S22

Enrichment efficiency (times)	BPA	Carbendazim	TMTD	Diquat	Anthracene
1	605.29	642.71	435.12	598.02	521.49
2	595.01	589.34	449.48	586.95	547.61
3	624.69	613.36	475.62	572.63	526.94
Average	608.33	615.14	453.41	585.87	532.01
RSD	2.48%	4.34%	4.53%	2.17%	2.59%

Supplementary Figure 23 | Enrichment efficiencies of five organic pollutants after enrichment process of MN-PCDP.

Supplementary Figure 24 | a Optical photograph and b bright-field optical images of analyte and Au nanoparticles on the hydrophobic slippery SERS platform.

Supplementary Figure 25 | Enhanced Raman spectra of TMTD (1 pM) in five independent batches after the enrichment of MN-PCDP measured by hydrophobic slippery SERS platform.

Supplementary Figure 26 | Enhanced Raman spectra of TMTD (0.5 pM) in five independent batches after the enrichment of MN-PCDP measured by hydrophobic slippery SERS platform.

Supplementary Figure 27 | Enhanced Raman spectra of TMTD (0.1 pM) in five independent batches after the enrichment of MN-PCDP measured by hydrophobic slippery SERS platform.

Supplementary Figure 28 | Enhanced Raman spectra of TMTD (50 fM) in five independent batches after the enrichment of MN-PCDP measured by hydrophobic slippery SERS platform.

Supplementary Figure 29 | Enhanced Raman spectra of TMTD (10 fM) in five independent batches after the enrichment of MN-PCDP measured by hydrophobic slippery SERS platform.

Supplementary Figure 30 | Enhanced Raman spectra of TMTD (5 fM) in five independent batches after the enrichment of MN-PCDP measured by hydrophobic slippery SERS platform.

Supplementary Figure 31 | Enhanced Raman spectra of TMTD **a** before and **b** after the enrichment of MN-PCDP adsorbent measured by aggregating approach with Au colloid. The characteristic peaks of TMTD were at 558 and 1379 cm^{-1} .

Supplementary Figure 32 | Enhanced Raman spectra of TMTD with **a** 1 nM, **b** 0.1 nM, **c** 10 pM and **d** 1 pM in five independent batches after the enrichment of MN-PCDP adsorbent measured by aggregating approach with Au colloid.

Fig. 5 e Raman spectra about selective enrichment of organic pollutant molecules (carbendazim, 1 nM) from practical samples.

3. In my opinion, a SERS nanosensor that combines ultrahigh sensitivity (achieved by high enrichment ability and signal enhancement ability), high spectral reliability, and fast removal speed, is publishable in the high-level journal of Nature Communication.

Answer: Thanks for the comment and very good suggestions from the reviewer. Following the suggestion, the revised manuscript has been obviously improved. After revision, we think the manuscript realize the review's criteria for publishing in the high-level journal. In this work, we show an enrichment-typed sensing strategy by using a powerful mesoporous nanosponge. Based on the excellent capturing selectivity of and fast removal capability for organic micropollutants from β -cyclodextrin polymers as well as the magnetic nanoparticles, great enrichment and detection performance on analyte are achieved, e.g. $\sim 90\%$ removal efficiency (RSD $<1\%$) within ~ 1 min, concentrated and enriched from complex matrix with an enrichment factor up to $\sim 10^3$ (RSD $<5\%$), ultrahigh detection sensitivity (0.5 pM with 100% detection possibility and 5 fM with $\sim 10\%$ detection possibility for TMTD molecule). By means of the current absorption strategy, the mesoporous nanosponge is used to separate and selectively enrich (2 \sim 3 orders of magnitude) target molecules from the real-sample system. Importantly, the current enrichment strategy is proved to be helpful tool in a variety of fields for portable and fast detection, such as UV-vis, Raman and fluorescent. Therefore, we believe this revised version with many supplemented data can be considered and accepted for publication.